# Real-Time MRI-Guided Prostate Interventions

**DOI:** 10.3390/cancers14081860

**Published:** 2022-04-07

**Authors:** Seyedeh Nina Masoom, Karthik M. Sundaram, Pejman Ghanouni, Jurgen Fütterer, Aytekin Oto, Raj Ayyagari, Preston Sprenkle, Jeffrey Weinreb, Sandeep Arora

**Affiliations:** 1Department of Radiology, Hospital of University of Pennsylvania, Philadelphia, PA 19104, USA; seyedehnina.masoom@pennmedicine.upenn.edu (S.N.M.); karthik.sundaram@pennmedicine.upenn.edu (K.M.S.); 2Department of Radiology, Stanford University Medical Center, Palo Alto, CA 04304, USA; ghanouni@stanford.edu; 3Department of Radiology, Radboud University Nijmegen Medical Center, 6525 GA Nijmegan, The Netherlands; jurgen.futterer@radboudumc.nl; 4Department of Radiology, The University of Chicago, Chicago, IL 60637, USA; aoto@radiology.bsd.uchicago.edu; 5Department of Radiology and Biomedical Imaging, Yale School of Medicine, New Haven, CT 06510, USA; raj.ayyagari@yale.edu (R.A.); jeffrey.weinreb@yale.edu (J.W.); 6Department of Urology, Yale School of Medicine, New Haven, CT 06510, USA; preston.sprenkle@yale.edu

**Keywords:** multiparametric magnetic resonance imaging (mpMRI), prostate cancer (PCa), prostate biopsy, prostate ablation, MRI-guided prostate interventions

## Abstract

**Simple Summary:**

Magnetic resonance imaging has shown to be a reliable imaging method for detecting clinically significant prostate cancer and directly targeting lesions during biopsy. As newer treatment methods emerge, the role of MRI in minimally-invasive (focal) treatment of prostate cancer is also increasing. Here, we review the real-time MRI-guided prostate interventions for prostate cancer diagnosis and treatment, focusing on the technical aspects of each modality.

**Abstract:**

Prostate cancer (PCa) is the second most common cause of cancer death in males. Targeting MRI-visible lesions has led to an overall increase in the detection of clinically significant PCa compared to the prior practice of random ultrasound-guided biopsy of the prostate. Additionally, advances in MRI-guided minimally invasive focal treatments are providing new options for patients with PCa. This review summarizes the currently utilized real-time MRI-guided interventions for PCa diagnosis and treatment.

## 1. Introduction

Prostate Cancer (PCa) is the most common non-cutaneous cancer and the second most common cause of cancer deaths in American males. It affects millions of men globally (GOLOBCAN 2020, [1]). An aging population, the increased prevalence of prostate-specific antigen (PSA) screening, and a lowered PSA threshold to recommend biopsy have contributed to an increasing incidence of PCa [2,3]. Increasing data now suggest a limited benefit of invasive surgery for low-risk localized PCa (Gleason score ≤ 6; ISUP 1) [4,5]. On the other hand, the treatment of intermediate/high-grade PCa with the definite whole-gland treatments, including radical prostatectomy and radiation therapy, can lead to marked comorbidities, such as incontinency and sexual dysfunction, with a significant impact on the patient’s quality of life [6]. Thus, there is a rising demand for minimally invasive techniques that are reliable for detecting and effective for treating clinically significant PCa.

This article will review the role of magnetic resonance imaging (MRI) in detecting clinically significant PCa and directly targeting lesions during biopsy. It will also discuss the newest minimally invasive interventions utilizing interventional MRI methods in guiding localized PCa therapy.

## 2. Prostate Cancer Diagnosis

### 2.1. Role of MRI in Prostate Cancer Detection

Multiparametric MRI or mpMRI combines the use of anatomic data and functional information from different pulse sequences for the detection, localization, and local staging of PCa. For example, T2 weighted imaging (T2WI) evaluates the anatomy of the prostate gland by providing contrast between tissues mainly based on the difference in their water and fat content. Visible PCa in the central gland and peripheral zone usually demonstrates abnormalities on T2WI. Additional sequences within an mpMRI protocol include diffusion weighted imaging (DWI), which detects the restricted motion of water molecules that often occurs in cancer, and pre- and post-contrast T1-weighted imaging (T1WI) with gadolinium-based contrast agents, which detects vivid and early enhancement within PCa.

### 2.2. Biopsy

In recent years, diagnostic pathways focusing on targeted biopsy of MRI-visible lesions suspicious for PCa have increasingly supplanted the prior standard practice of 12-core systematic ultrasound (US)-guided biopsy of the prostate [7]. Targeting MRI-visible lesions has been beneficial and led to an overall increase in the detection of clinically significant PCa and a decrease in the detection of clinically insignificant PCa [8,9].

MRI targeted biopsy can be done via three techniques:Cognitive biopsy: MRI images are reviewed prior to performing and during ultrasound (US)-guided biopsy to visually estimate the location of the MRI abnormality.MRI–US software fusion biopsy: Obtaining diagnostic MRI images and fusing them with real-time US images during biopsy.“In-Bore” real-time MRI biopsy: Biopsy done with the patient in the MRI scanner.

Each targeted biopsy technique has advantages and disadvantages (Table 1). The optimal technique is yet to be determined, but fusion biopsy or in-bore biopsy are usually preferred over cognitive biopsy, which suffers from key disadvantages, such as operator dependence, an absence of standardization, the inability to confirm needle location in the target lesion, and a reduced effectiveness for small lesions [10]. Some recent studies suggest a better performance of in-bore real-time MRI biopsy versus fusion biopsy [11,12], but prospective trials are still needed. In-bore biopsy is less available, more challenging to implement, more time-consuming, and more difficult to perform a concurrent systemic biopsy. Since up to 10% of clinically significant PCa is not detected on prostate MRI, the systemic biopsy afforded by the other approaches may detect this 10% of PCa. In some institutions, in-bore biopsy may be reserved for discrepant or difficult cases.

In-bore biopsy can be performed via either transrectal, transperineal, or rarely, transgluteal approaches. The methodology of transrectal and transperineal approaches will be discussed, focusing on the crucial technical steps. The transrectal approach is more commonly used but has a higher risk of sepsis and is also associated with hematochezia [13]. The transperineal approach is sometimes favored due to the lower risk of sepsis, but it is associated with a higher rate of urinary retention. Some institutions give IV antibiotics prior to transrectal biopsy to reduce the risk of sepsis.

In-Bore Transrectal: Commercial platforms are available. Patients are usually placed prone. A needle sleeve of the biopsy system, which serves as a guide and a fiducial marker, is inserted into the rectum. This sleeve is attached to a clamp stand that attaches to the MRI tabletop. T2W images are then obtained, and registration of the needle sleeve is done. The center of the target lesion is identified, after which the software calculates the necessary mechanical adjustments needed for the needle sleeve to align with the desired trajectory. The biopsy needle is then placed in the needle sleeve, and the correct location is confirmed. After the needle is fired, imaging is repeated to confirm the position [14].In-Bore Transperineal: Patients are usually placed supine with legs in an MRI-compatible stirrups device. A localizer grid is placed and secured at the perineum. The reservoir of the marking block is usually filled with water. T2W images are then obtained. The point of insertion in the marking block and the depth of the target lesion are then calculated, either using available software or manually. The marking block can be left in place during the entire procedure, but some institutions remove it after identifying the skin entry site [14]. The lack of the mark block allows for fine adjustments to the needle path if needed. Images are obtained as the needle is advanced into the target, while making adjustments as needed (Figure 1).

**Figure 1 cancers-14-01860-f001:**
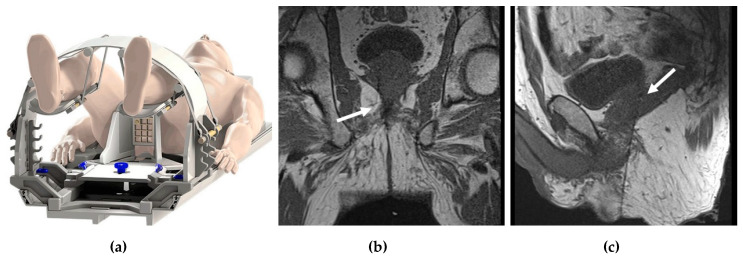
In-bore transperineal biopsy: (**a**) diagram of a patient placed supine in an MRI-compatible stirrups device and grid; (**b**) coronal and (**c**) sagittal T1-weighted MRI images demonstrating the needle entering the right prostate gland (white arrows).

An office-based, low-field MRI system for prostate biopsy has recently been granted 510 (K) clearance by the FDA. This innovative MRI system operates between 58 mT and 74 mT and is intended to be used for transperineal prostate biopsies and treatments. The scanner does not require any special room modifications or a dedicated certified MRI technologist. It has a limited fringe field and low energy requirements, and no cryogens are needed. The scanner is single-sided, i.e., there is no “bore to place a patient”. It projects the magnetic field in front of the scanner, and the patient is positioned in a lithotomy position with the prostate in the center of the field of view (Figure 2). To guide biopsies, the images obtained by this system are fused with the previously obtained high-resolution T2-weighted images acquired as part of a diagnostic mpMRI per PIRADS V2.1 (Figure 2d). A native biopsy planning software and a needle placement grid are utilized for the determination of the site of the biopsy. Unpublished phantom and limited patient studies using this system demonstrated acceptable navigation and target registration error, along with submillimeter patient motion. This technology can potentially reduce MR–US misregistration errors and be independent of the scheduling pressures on clinical MRI imaging systems. Other low-field MRI systems are in development.

## 3. Prostate Cancer Treatment

Long-term data from the ProtecT trial have shown that, at a median of 10 years, PCa-specific mortality was low for localized PCa, irrespective of the treatment assigned (surgery, radiotherapy, or active surveillance), with no significant difference among treatments [15]. However, surgery and radiotherapy were associated with lower incidences of disease progression and metastases (20%) than was active surveillance. Given that radical whole-gland treatment with surgery or radiotherapy is associated with significant morbidity, including urinary incontinence and erectile dysfunction, therapeutic options that can offer lower incidence of disease progression and metastases than active surveillance, while avoiding morbidity associated with surgery and radiation, may be attractive for many patients.

Towards this goal, a wide range of minimally invasive modalities are being evaluated for whole-gland, partial-gland, and focal treatment of PCa, including focal laser ablation, cryoablation, irreversible electroporation, and high-intensity ultrasound (sonoablation). Table 2. summarizes the advantages and disadvantages of MR- vs. US-imaging-guided prostate ablation techniques.

### 3.1. MRI vs. Ultrasound for Guidance of Prostate Ablation

MRI allows for a precise, multiplanar, and real-time needle guidance in contradistinction to US, which is usually uniplanar.

Ultrasound guidance needs an intact non-diseased rectum, as the imaging is done transrectally, and needles may be placed transrectally or transperineally. Using MRI, interventional devices can be placed transurethrally, transgluteally, transperineally, or transrectally. MRI guidance can be used in patients without a rectum (Figure 3a).

Under MRI guidance, widening of the rectoprostatic space is possible to protect the rectum and neurovascular bundle (NVB) [16]. US guidance is usually done by a transrectal transducer; the interventional needle may be inserted under transrectal or transperineal US visualization. Any widening of the rectoprostatic space under ultrasound will decrease image quality and needle visualization.

During ultrasound-guided cryoablation, there is poor visualization of the ablated zone due to the angle-shadowing effect of the ice ball [17]. In contrast, MRI offers artifact-free visualization of the ice ball (see discussion below).

#### 3.1.1. General MRI-Guided Prostate Ablation Procedure Precautions

MR thermometry measures increases in temperature due to heat application. The base temperature is measured at the core body temperature. As the typical reduction in body temperature under general anesthesia is compounded by the relatively cool ambient room temperature in MRI facilities, a patient’s core temperature can drift below 35 °C by the time treatment is initiated. To prevent hypothermia, thereby reducing anesthesia time, we recommend providing the patient with proper insulation and maintaining a comfortable MRI room temperature.Pressure sore prevention: Unfortunately, MRI tables have hard surfaces. During these procedures, time on the MRI table can be many hours compared to standard imaging MRI times, which rarely exceed 1 h. To minimize injury and potential pressure sores (especially in thin patients), extra padding is used. However, given the small diameters of MR bores, a balance between adequate padding while allowing enough room in the scanner for the patient and equipment must be achieved.Managing patient motion: In addition to peristalsis, other sources of patient motion that can cause imaging artifacts and inaccurate measurements of temperature during prostate treatment include bladder filling, breathing, and involuntary movement of the pelvic floor muscles. Treating radiologists should correctly identify the patient motion and temperature measurement artifact to stop the sonication and replan before reinitiating treatment delivery. To prevent patient and muscle movement during treatment delivery, a neuromuscular blockade (e.g., rocuronium) should be supplemented before treatment initiation. Bowel and bladder motion reduction strategies are also employed.Adequate treatment margins in focal therapy: Most reported uses of ExAblate utilize an ablative margin of at least 5 mm, which has seen promising results in many studies [18,19]. However, some authors raise the possibility that MR imaging underestimates the true margins of index lesions by 50% or more, depending on the sequences [20]; therefore, ablative therapies may undertreat. For example, a patient with recurrent in-field PCa on subsequent biopsy, which was ultimately treated with uncomplicated prostatectomy, might have been the result of a marginal lesion not being included within the 5 mm prescribed margin. The patient might have had a different outcome if the margins were increased to 9 mm. Due to limited studies (e.g., where the patient underwent radical prostatectomy after treatment), this idea requires additional study and validation. Ultimately, the margins should be decided by the treating physician, and further, a radiologist with experience in interpreting prostate MRI by PI-RADs should be involved in treatment planning and execution. The trade-off for increasing ablative margins is a higher prescribed volume and longer treatment time. Treatment margins of 15 mm have also been theorized [21]. There may also be instances where the full margins may not be fulfilled, secondary to adjacent non-target anatomy.Rectal gas can be problematic during MRI-guided procedures. It can cause imaging artifacts due to material interphases. Additionally, rectal gas can also cause reflection or absorption of the ultrasound beam, leading to unintended heating along the rectal wall as well as misregistration of thermographic data. Rectal gas reduction strategies include diet control prior to procedure, bowel preps, and device manipulation in case of gas trapping during the procedure.

#### 3.1.2. Transurethral MRI-Guided High-Intensity Focused Ultrasound (HIFU)

Transurethral ultrasound ablation (TULSA) is a completely noninvasive option that allows the transmission of mechanical energy (ultrasound waves) through intact skin or mucosa toward targeted tissues. The energy is converted to thermal energy that produces temperature increases causing the coagulative necrosis of benign and malignant tissue. TULSA can be used for whole-gland, partial-gland, or focal ablation and is done under general anesthesia.

The TULSA-PRO™ system incorporates an US transducer, robotic positioning system, and an endorectal cooling device. The US transducer is a 22 Fr (7.5 mm) rigid tube with a soft coudé tip. The robotic positioning system provides linear and rotational motion of the applicator in the prostatic urethra. The linear array incorporates 10 individual transducers of 5 mm length. The transducers emit high-intensity ultrasound directly into the adjacent prostate tissue. Cooled fluid flowing through the transducer (sterile water) and the endorectal cooling device (water with manganese chloride and surfactant) protect 1–2 mm of periurethral tissue and the rectal wall from thermal damage, respectively.

The procedure can be performed in a 1.5 T or 3 T MRI scanner (Phillips or Siemens) using a 32-channel anterior/posterior cardiac coil array. MRI thermometry is performed with an EPI that is temperature-sensitive, using a proton-resonance frequency shift-induced phase difference sequence acquired in 12 axial slices of 4 to 5 mm thickness and in-plane resolution of 2 mm. The thermometry data are refreshed every 5 to 7 s, with an accuracy ≤1 °C and a precision ≤2 °C. Directional therapy is displayed on thermometry as a “flame” shape. Thermometry feedback maps utilize a feedback algorithm to adjust the transducer’s rotation and each element’s frequency/power. The base of the flame is hottest and decreases to 57 °C towards the ablation edge. The algorithm prescribes an ablation volume to reach a temperature of 57 °C within 2 mm of the prostate capsule boundary, translating to 240 CEM43 lethal thermal margins beyond the capsule. A Tmax is set at 87 °C to prevent progression to tissue boiling and resultant internal cavitation. After the treatment of an angular sector, the robotic positioning system algorithmically rotates the transducer through each subsequent angular sector to complete the ablation volume. The maximum treatable distance of this device is 3 cm, i.e., the outer margin of the lesion should be no more than 3 cm from the center of the urethra with the UA in place [22].

After device insertion and registration under MRI guidance, the ablation volume is defined utilizing high-resolution 2D T2W imaging with MR thermometry, obtained perpendicular to the transducer and centered across each individual element to completely cover MR-visible lesions. The prostate boundary within the ablation volume is drawn to avoid the neurovasculature, the rectal wall, and the bladder neck/urethral sphincter. These individual 2D boundaries across the 10 transducer elements equates to a 3D ablation volume. Transducer elements outside the intended treatment area are simply rendered inactive. The first ablation cycle starts a few degrees (clockwise or counterclockwise) before the visible index lesion or presumed location based on biopsy. Continuous MR thermometry during treatment provides closed-loop feedback control. After completing treatment, post-contrast axial T1-weighted images provide an assessment of the acute nonperfused volume (NPV) (Figure 3d).

#### 3.1.3. Transrectal MRI-Guided High-Intensity Focused Ultrasound (HIFU)

The ExAblate 2100 prostate device (INSIGHTEC, Haifa, Israel) is a CE-approved transrectal MRI-guided device utilized for the focal treatment of index lesions in PCa. ExAblate is only intended for focal lesion therapy, rather than a whole-gland approach. The system is a beam-steerable endorectal phased-array transducer containing 990 elements and operating at 2.3 MHz and up to 30 W. The transducer uses a single-use balloon containing degassed water at 14 °C to provide protective cooling for rectal tissue. The ExAblate system is integrated with GE MRI scanners and provides real-time thermometry using PRF of the target area.

Prior to initializing sonication, T2WI (Figure 4a) and DWI (not shown) are acquired. After identifying and contour-mapping the treatment zone, the integrated treatment software develops a therapeutic plan (Figure 4b). This plan includes energy levels as well as sonication numbers, size, shape, and overlap. Typical treatment sizes are cylindrical and measure 2 mm in diameter by 8 mm in anterior–posterior dimension, often analogized in size to a “grain of rice” (Figure 4b). Tumor margins are initially set at 5 mm, but are user-adjustable to account for sensitive, non-target structures, such as the urethra, urethral sphincter, bladder wall, rectal wall, and neurovascular bundles. Pretreatment microsonications are performed to confirm the appropriate treatment fields with proton-resonance MR thermometry prior to therapeutic macrosonication. Real-time thermography monitors for treatment completion at Tmax 65 °C (Figure 4c). Focal re-treatment may be performed if the thermography suggests an area of undertreatment.

Patients are positioned in a lithotomy position, and a Foley catheter is placed. Most cases are performed utilizing general anesthesia; however, some studies have also reported the use of regional anesthesia. Patient selection has been suggested to be limited to patients with a prostate gland volume of 70 mL and a maximum lesion distance of 6 cm from the transducer. Index PCa should be identified either by pre-procedural biopsy or visualization of the lesion by MRI.

Post-procedurally, a contrast-enhanced MRI is performed to visualize the non-perfused volume and to confirm treatment completion (Figure 4d).

#### 3.1.4. MRI-Guided Focal Laser Ablation

Focal laser ablation involves the deposition of laser radiation energy within a target lesion, causing rapid temperature elevation and tissue destruction. The Visualase laser ablation system is one of the most commonly used systems for MRI-guided in-bore prostate laser ablation (FDA-approved in 2007). This can be used with 1.5 or 3 T systems. The system is comprised of a 600-μm fiber optic cable with a distal diffusing tip for dispersing 980 nm of laser light generated from a 30-watt diode laser. The fiber optic cable is contained within a 1.85 mm polycarbonate cooling cannula. The cooling cannula is connected to a saline pump to prevent overheating and charring adjacent to the fiber during the ablation. The laser-diffusing fiber tip creates a 16- to 18-mm oval ablation zone, and the laser can be advanced or withdrawn within the cooling cannula to “paint” an ablation zone.

This ablation system can be used with both transperineal and transrectal (Figure 5) approaches with their associated platforms. MR–US fusion guided laser ablation has also shown to be possible using temperature probes for monitoring, under local anesthesia. The procedure is usually done under general anesthesia with a bladder catheter in place. Saline and lidocaine can be injected in the rectoprostatic space for potentially improved protection of the neurovascular bundle and the rectum [23].

In the beginning of the procedure, the cooling cannula is advanced into the targeted portion of the prostate under MRI guidance, and the stiffener (which is made of titanium) is removed. Subsequently, the laser fiber optic cable is inserted. Image co-registration with the MR thermometry images is then done. The dose is monitored using proprietary MR thermometry software, and the laser fiber is visualized as a linear artifact [24]. Initially, a sublethal low-power test is performed to confirm the appropriate fiber placement. After any needed adjustments are made, the laser power is increased to a lethal dose, and the ablation zone is monitored continuously. The general values of ablation wattage range from 15 to 20 W over an ablation time of 2–4 min. The minimal treatment temperature is 60 degrees Celsius. The laser fiber can be advanced or withdrawn in the cooling cannula to create multiple overlapping zones of ablation to ensure an adequate treatment margin.

There were concerns about overheating, thermal damage, and the accuracy of MR thermometry, which led to the recall of these devices. FDA guidance provides mitigation strategies for overheating:Keep the temperature below 90 degrees right outside or immediately adjacent to the laser fiber’s image artifact. This action prevents excessive heat near the fiber from charring tissue or damaging the cooling cannula. Carbonization or charring of the tissue near the laser fiber prevents the effective transfer of heat to more peripheral tissue, thereby limiting the effective size of the ablation zone. If the temperature exceeds the set limit, there is an automatic shutoff.Critical structures such as NVB are set to low temperature targets of 43 °C or less. If the temperature exceeds the set limits, there is an automatic shutoff.Consider heating the target tissue slowly to reduce the potential for inaccurate MR thermometry readings. Additionally, heating the tissue slowly may lessen unanticipated thermal spread.Keep the cooling system running throughout thermal monitoring, including when the laser is on and after it is shut off, to bring the tissue next to the fiber back to the baseline temperature within 120 s after laser delivery.

### 3.2. General MRI-Guided Thermal Therapy Monitoring

Proton-resonance temperature mapping (PRF) utilizes the phenomenon of the linear change of resonance frequency of water protons with temperature. However, PRF has three major limitations [25]:PRF temperature mapping is highly sensitive to motion and tissue edge artifacts. Initially, a baseline comparison image is obtained, and all subsequent images are compared to it. As a result, any minor motion of the patient, bowel, or bladder can disrupt the baseline image alignment, causing phase registration artifacts. Reference-less temperature mapping has been proposed to alleviate this.In postsurgical prostate beds and in prostates treated with brachytherapy, the presence of the surgical clips can cause significant metallic artifacts, essentially making phase-change-based temperature imaging nondiagnostic.Measuring temperature at the fat-water interphase is problematic for PRF thermometry, as it is only dependent on the temperature of water protons. The resonance frequency of protons in fat is different, which produces artifacts and temperature measurement inaccuracy for tissue–fat interfaces. Some approaches have attempted to resolve this by using Dixon fat-water separation techniques, using the PRF method on the fat-only images, and using the phase changes of the fat signal to correct for non-temperature-dependent phase changes.

For confirmation of the ablation zone, a contrast-enhanced T1W scan is usually done (with optional subtraction images) at the end of treatment. Re-treatment after gadolinium contrast administration is not performed, as the contrast agents can be retained in the ablated tissue for several hours [26]. This makes the evaluation after re-treatment suboptimal, and causes potential toxicity if the stability of the gadolinium chelate is compromised by markedly increased tissue temperatures. T2*-weighted MR images with long TE have been shown to visualize the post-procedure focal laser ablation zone comparably to the contrast-enhanced T1-weighted MRI. Other sequences, such as diffusion-weighted imaging, have also been utilized for thermal therapy monitoring [27].

In both therapeutic US modalities, calcifications cause a high percentage of energy to be locally absorbed or reflected, with little transmission. This can cause an excessive heating of the tissue between the transducer and calcification, and a suboptimal heating beyond the calcifications. Additionally, calcifications cause image artifacts on the axial dynamic segmented-EPI sequence used for the MR temperature mapping, due to a decreased signal intensity within the calcification and signal dephasing in the surrounding soft tissues. This results in a locally increased temporal standard deviation of MR temperature measurements (increased temperature mapping uncertainty). The presence of calcifications in the treatment beam path may be a contraindication to therapeutic US depending on their location and size. For therapeutic US, the presence of large cysts in the beam path may also be a contraindication.

### 3.3. MRI-Guided Cryoablation

Cryoablation is a technique that uses freeze–thaw cycles to produce tissue destruction and necrosis [28]. MRI-guided cryoablation is usually performed under general anesthesia. The widening of the rectoprostatic space may be performed by placing a needle and infusing saline/lidocaine to improve rectal and external urinary sphincter protection. Multiple probes are placed within the prostate via a transperineal approach under MR or ultrasound guidance using a grid approach [17]. The number of probes depends on focal, hemi-, or whole-gland ablation. Cryoprobes should be no more than 2.0 cm apart, and there should be no less than 0.8 cm between a cryoprobe and the urethra [29]. Argon gas is funneled through the probe needles to freeze the target within the prostate, and helium gas is used in the thawing phase. A transurethral warming catheter is usually placed to provide protection against urethral damage.

Monitoring during MRI-guided cryoablation is usually done by visualizing the growth of the ice ball. Due to the ultrashort T1 and T2 times of the solid ice ball, it is well-visualized compared to untreated tissues on an MRI. Temperatures of −40 °C reached on at least two successive freeze–thaw cycles ensure the ablation of the most tissue [25]. The ice ball apparent to the operator on MRI can be larger than the oncologically effective ablation zone. Ice ball margins usually record a temperature of −0 °C. However, the lethal freezing temperature for cancer tissue is around −40 °C, and the lethal freezing margins may actually be situated approximately 1 cm within the ice-ball edge, suggesting that the ice ball should extend at least 1 cm beyond the identified treatment zone. Some reports have also suggested that the actual ablation margin is best demonstrated with contrast enhancement post-procedure and is actually at the −20 °C isotherm [25]. Ensuring sufficient margins is challenging within the small space of the male pelvis, where critical structures such as the rectum, the external urethral sphincter, and the neurovascular bundle need to be monitored and spared; potential protective steps for these structures are described above. In addition to visualizing the ice ball, MR thermometry can also be applied to cryoablation techniques, using ultrashort echo times in order to ensure optimal tumor destruction. MR-compatible thermocouples are also routinely used for temperature monitoring (Figure 6).

Cryoablation is FDA-approved with varying insurance reimbursement amounts. AUA guidelines include a conditional recommendation for whole-gland cryoablation for the treatment of intermediate-risk prostate cancer.

### 3.4. Other MRI-Guided Prostate Cancer Treatment Procedures

Irreversible electroporation for the focal treatment of PCa can be performed under MRI guidance, but it is usually done under US guidance in a vast majority of patients [30]. Similar to other ablation techniques, whole-gland, hemi-gland, or focal ablations can be performed. The procedure uses short electrical pulses to effectively destroy cancer cells. An MR–TRUS fusion technique can be utilized for transperineal electrode placement using a mapping biopsy/brachytherapy template. The ablation zone extends between each pair of individual electrodes and 5–10 mm around the electrodes. The distance between individual electrodes is between 0–20 mm. After satisfactory electrode placement into the prostate, the electrodes are then connected to the IRE generator. The IRE protocol may include a total of 90 pulses, with a pulse length of 70 μs, to achieve a current flow of 20–50 A between each electrode pair [31]. Intraprocedural imaging monitoring is rarely utilized. Pre-and post-procedure imaging can be done using contrast-enhanced MRI and CEUS.Photodynamic therapy (PDT) involves the activation of a photosensitizer agent by a specific wavelength of light and in the presence of oxygen, leading to the production of reactive oxygen species (ROS) [32]. This in turn leads to cell necrosis and death. Along with IRE, cryoablation, and laser ablation, PDT can be done under US-guidance using MR–US fusion as a focal therapy strategy. The goal of PDT is to selectively damage PCa while sparing the neurovascular bundles, sphincter, and urethra and limiting toxicity. During PDT for the treatment of PCa, photosensitizing agents are delivered orally or intravenously and allowed to localize to vasculature or to the tumor site. On treatment, low-powered laser light is delivered by an optical fiber inserted into the suspected PCa lesion (transperineally) under MR–US fusion guidance. Upon the absorption of laser energy, the accumulated photosensitizing agents at the targeted site create ROS and cause cell death. The laser power is non-ablative, and no intraprocedural monitoring is needed. Pre- and post-procedure imaging can be done using contrast-enhanced MRI or CEUS.Multiple trials have been performed with different photosensitizing agents [33]. Some agents are administered and allowed to accumulate at tumor sites over several hours or days prior to treatment. Other PS agents target vasculature, and treatment is delivered within minutes after injection. Compared to active surveillance, PDT has been demonstrated to be a safe and effective tissue-preserving approach for low-risk localized PCa [34]. PDT can be used as a salvage therapy after radiation therapy as well.MRI can also be incorporated in the prostate brachytherapy procedure on different levels: real-time brachytherapy seed implantation guidance, high-dose-rate (HDR) dose optimization, and low-dose-rate (LDR) post-implant dosimetry [35]. Table 3. compares various image-guided focal treatment strategies for summary.

### 3.5. Focal Treatment Complications and Related Factors

Common complications after focal PCa therapies include hematuria and urinary infection, and complications related to the indwelling Foley catheter insertion, such as discomfort and urethral sloughing. The most common complications after focal therapy usually occur within the first 30 days after the intervention [36]. The pre-operative estimation of the size of the prostate is vital. Exceptionally large prostates may not be candidates for some energy sources or treatment planning, such as TULSA. These patients are at an increased risk for developing significant lower urinary tract symptoms (LUTS) after treatment. The exact location of the cancerous tissue is also a key factor in pre-operative planning, which can predict the type and frequency of complications. Postoperative obstructive LUTS and irritation are likely to occur in PCa located near the urethra or the bladder neck. PCa located close to the neurovascular bundles with capsular contact require an extended ablation, which could have an impact on erectile function recovery [37].

## 4. Discussion

MR imaging has markedly improved primary PCa visualization. This has led to a change in the paradigm of PCa diagnosis from random prostate biopsies to targeted/combined biopsies. The suspected lesion seen on MRI can be targeted using US guidance via cognitive needle placement or actual MR–US fusion, but an in-bore biopsy using real-time MRI guidance is arguably the most accurate, especially for smaller lesions. However, given the ease of operation and relatively good performance, MR–US fusion biopsy has become the dominant diagnostic modality for PCa diagnosis. In-bore biopsy may be a favored modality in some centers with specialized expertise and utilized to troubleshoot cases in which there is radiologic–pathologic discordance after MR–US or cognitive fusion biopsy. The development of portable low-field MR scanners and high-resolution ultrasound imaging may challenge these paradigms in the future.

Urinary, sexual, and gastrointestinal side effects of the traditional treatments of localized PCa have led to the development of many minimally invasive prostate cancer treatment modalities, the most common being laser, cryo-, and sonoablation. MRI guidance for treatment offers distinct advantages over US-guidance as described (Table 2). These minimally invasive modalities can be used to treat PCa using whole-gland, partial-gland, or focal therapy treatment approaches. Prospective data for the superiority of one of these modalities over the others are lacking. Each of these has unique advantages and disadvantages. Some practitioners suggest an a la carte approach of matching the treatment modality based on tumor location [37]. The discussion of the oncologic outcomes of minimally invasive prostate treatments vis-à-vis traditional approaches is beyond the scope of this article. No prospective comparative randomized controlled trials, and long-term efficacy data are available for the newer technologies, such as MRI-guided US ablation. Nonetheless, these are slowly becoming attractive treatment options, mostly due to their excellent side effect profiles, especially when used in the context of focal therapy. Still, more studies are needed to determine the exact place of these technologies in the pantheon of localized PCa treatment options.

Further developments in this field may include faster MRI sequences, additional low-field magnets and capabilities for MRI-guided interventions, and PET/MRI targeted interventions. Additional research is necessary for the detection of PCa recurrence, especially by mpMRI, as the procedural changes in the prostate can mask recurrent disease.

## 5. Conclusions

The use of MRI for PCa interventions is the natural progression of the ability of MRI to accurately visualize clinically significant PCa lesions. Real-time MRI monitoring of PCa interventions is highly accurate and offers very tangible advantages over US-guidance. The main disadvantages of MRI guidance for prostate procedures are related to the ease of use, procedure time, availability, specialized equipment, and cost, most of which can be mitigated with experience and dedicated interventional equipment.

## Figures and Tables

**Figure 2 cancers-14-01860-f002:**
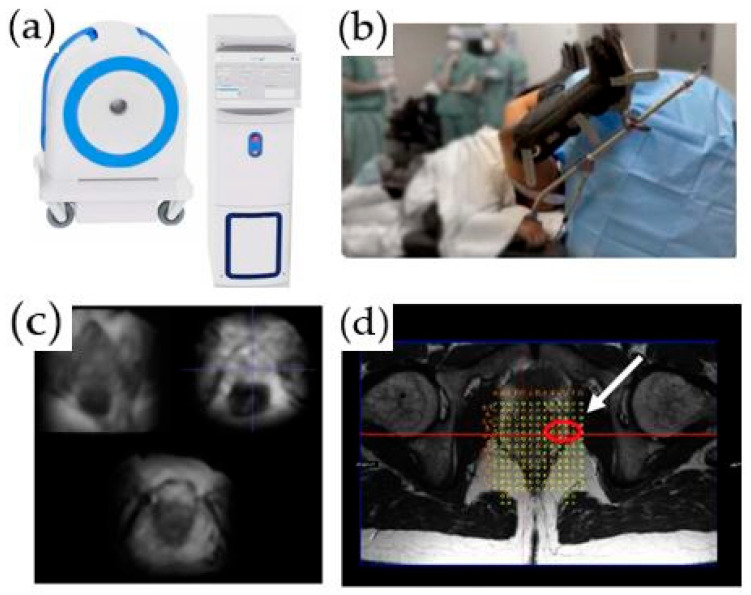
An office-based, low-field MRI system for prostate biopsy: (**a**) low-field MRI device; (**b**) biopsy device; (**c**) axial T2-weighted images acquired on the low-field magnet; (**d**) registered high-resolution axial T2-weighted image and grid for biopsy planning. A target is identified in the left lateral prostate (white arrow).

**Figure 3 cancers-14-01860-f003:**
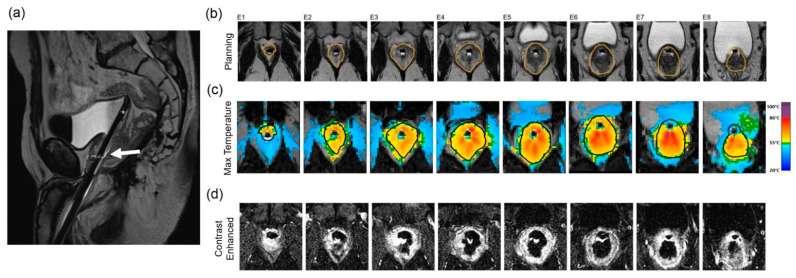
Treatment of a patient with localized, low-stage PCa with a TULSA-PRO device: (**a**) Sagittal T2-weighted MR image demonstrates the transurethral sonoablation device within the prostatic urethra and the tip within the bladder of a patient without a rectum (white arrow). (**b**) Axial T2-weighted MR planning images of the prostate at the level of 8 ultrasound transducers. The prostate boundary is contoured in orange. (**c**) MR thermometry images and color overlay demonstrate areas of heating. (**d**) Post-treatment imaging of the prostate acquired after gadolinium administration. Areas of bright signal demonstrate viable tissue, while dark areas indicate treated tissue.

**Figure 4 cancers-14-01860-f004:**
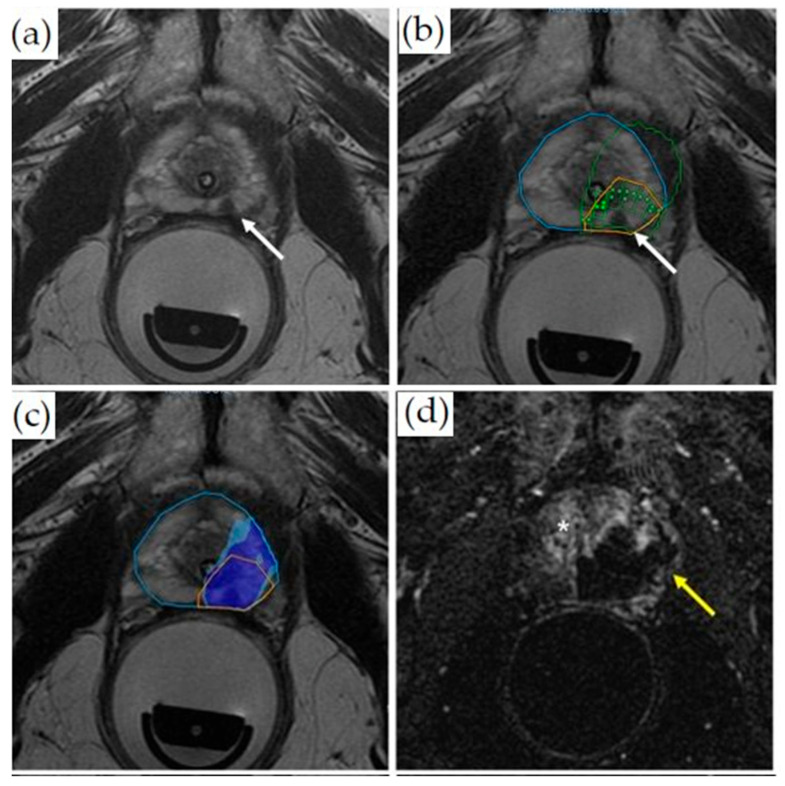
Treatment of a patient with Gleason 3 + 4 PCa with transrectal MR-guided HIFU (ExAblate 2100 device): (**a**) Pretreatment axial T2-weighted MR imaging demonstrates a hypointense lesion in the medial left peripheral zone (white arrow). (**b**) Overlayed treatment planning image shows the prostate contour (blue contour), region of interest with at least 5 mm margins (orange contour), focal spots (green dots inside rectangles), predicted thermal dose (green contour), and lesion (white arrow). (**c**) Thermal map overlay image demonstrates areas that have reached thermal dose based on intraoperative thermometry (dark blue at least 8000 CEM43, light blue 240 CEM43). (**d**) Axial post-contrast images obtained immediately after treatment demonstrate a non-perfused area (dark area, yellow arrow) indicating treatment. The viable prostate tissue demonstrates enhancement (bright signal, asterisk).

**Figure 5 cancers-14-01860-f005:**
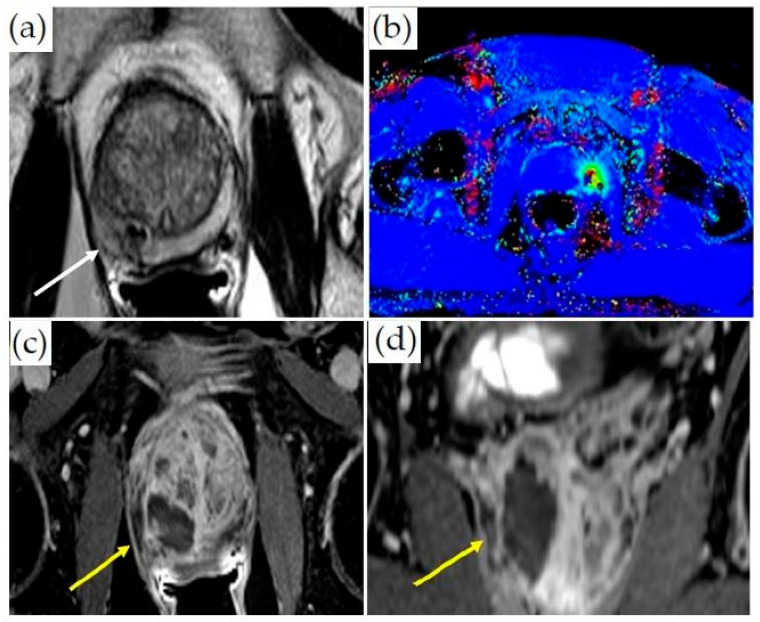
MRI-guided laser ablation: (**a**) Axial T2-weighted MR images with lesion in the right medial peripheral zone (white arrow). (**b**) Thermometry mapping (different patient) during treatment. (**c**) Axial and (**d**) coronal post-contrast T1-weighted imaging demonstrating ablation cavity (yellow arrows).

**Figure 6 cancers-14-01860-f006:**
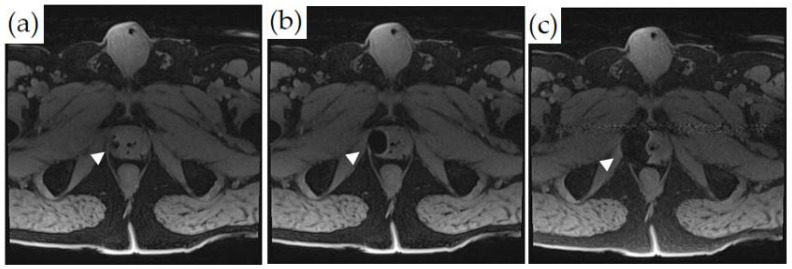
MRI-guided cryoablation and monitoring of ice-ball formation on axial T1-weighted fat-saturated images: (**a**–**c**) Ice-ball formation as a function of time in the right lateral peripheral zone of the prostate (white arrowheads). The untreated normal prostate tissue is visualized.

**Table 1 cancers-14-01860-t001:** Summary of advantages and disadvantages of different biopsy techniques.

Biopsy Technique	Advantages	Disadvantages
Cognitive	No additional equipment or software Least expensive Fastest Ease of obtaining additional systematic cores	Difficulty in targeting small lesions, US-invisible lesions More difficult localization in larger glands Operator dependent Calcifications can obscure target Inability to record biopsy site for future reference
MR–US Fusion	Ease of obtaining additional random systematic cores Recording position of target and systematic cores Can target US-invisible lesions Not affected by calcifications	Registration error between MR–US images Upfront acquisition costs Multiple steps prone to operator error/bias, including MRI gland and tumor segmentation, US gland segmentation, MR–US registration Learning curve Need for good collaboration between specialists
In-Bore	Arguably most accurate and least prone to errors Reduction in number of cores (if concurrent systematic cores not obtained)	Most expensive and time-consuming of the three techniques Limited availability Specialized MRI-compatible equipment More difficult to do concurrent systemic biopsy Can be ergonomically challenging

**Table 2 cancers-14-01860-t002:** Comparison of MR- vs. US-imaging guidance for prostate ablation.

	MR	US
Advantages	Greater spatial and contrast resolution. Real-time PRF MR thermometry to monitor temperature rise. Precise temperature monitoring is the most important difference and is being highlighted. Better visualization of tumor boundaries and peri-tumoral structures. Direct visualization of tumor. Complexity due to equipment and personnel.	Relative familiarity and ease of use. MR–US fusion available (however, small chance of registration). Ability to re-treat perfused areas after post-treatment microbubble-enhanced scans.
Disadvantages	Sensitivity to artifacts caused by motion and small air bubbles. Inability to re-treat after contrast administration. Hip prostheses/other metallic structures can preclude MRI guidance.	Secondary signs of temperature changes based on greyscale changes and RF pulse–echo backscatter are not true estimates of heat deposition. Usually, these secondary signs occur at higher temperatures, and temperature cannot be prospectively controlled.

**Table 3 cancers-14-01860-t003:** Overview of image-guided PCa focal treatment strategies.

Focal Treatment Technique	Main Principle and Energy Source	Imaging Modality	Approach
Photodynamic Therapy	Reactive oxygen species generated by the transfer of energy from the activated photosensitizing drug, causing apoptosis and cell death.	TRUS or MRI–TRUS fusion guidance	Transperineal insertion of laser fibers. Photosensitizing drug administered intravenously.
Laser Therapy	An optical laser fiber is placed within cancerous tissue by transrectal or transperineal approach. The energy transferred by this laser fiber raises the temperature of the targeted tissue above 60 °C, causing cell death.	MRI, TRUS, or MRI–TRUS fusion guidance	Transrectal or transperineal
Irreversible Electroporation	A non-thermal ablation technique. Electrical pulses traverse between transperineally inserted electrodes to produce irreversible cell membrane permeabilization, which causes apoptosis of the cells.	TRUS or MRI–TRUS fusion guidance	Transperineal
Cryoablation	Tissue ischemia and coagulative necrosis caused by alternative cycles of freezing and thawing of cancerous tissue, causing cell death.	MRI, TRUS, or MRI–TRUS fusion guidance	Transperineal
High-Intensity Focused Ultrasound	Energy from high-frequency ultrasound generates heat (>60 °C) at targeted tissues, leading to necrosis.	MRI, TRUS, or MRI–TRUS fusion guidance	Transrectal
Transurethral Ultrasound Ablation	The transurethral applicator provides a beam of focused energy to achieve a temperature of >55 °C, which induces thermal coagulation of the prostatic tissue.	MRI guidance	Transurethral
Radiofrequency Ablation	Tissue damage and coagulative necrosis caused by delivery of low-dose radiofrequency waves directly to the cancerous tissue.	TRUS or MRI–TRUS fusion guidance	Transperineal

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
