# Peer review of "Real-Time MRI-Guided Prostate Interventions"

_cancers, 2022, doi:10.3390/cancers14081860_

Round 1

Reviewer 1 Report

Comments;

Clinically useful review. Text and Tables are clear. Illustrations are mostly reasonable high quality.

  1. Page 2, line 59, “…detect enhancement…” can be changed to “…detect vivid and early enhancement within PCa…”.   
  2. “Transrectal approach is more commonly used but has a higher risk of sepsis and is also associated with hematochezia. Transperineal approach is increasingly becoming more popular due to lower risk of sepsis. It is, however, associated with a higher rate of urinary retention.” Line 84 to 87. We performed US/MRI fusion biopsy by transrectal approach with infection rate less than 0.5%, very similar to the rate of infection by transperineal approach, because we gave IV antibiotic before the biopsy.  I do not think transperineal approach is becoming more popular. 
  1. “In-bore Transrectal….” Line 88 to 96. Although the Table 1 mentioned the disadvantages of the method, more details of the disadvantages can be emphasized in this paragraph.  1, time consuming, 40 to 60 minutes per case as compared to the transrectal approach, 20 to 30 minutes. 2, You can not perform systematic prostate biopsy at the same time of the in-bore.  As we know, up to 10% clinically significant PCa will be missed by prostate MRI.  Imaging guided biopsy together with TRUS biopsy can pick up this 10% PCa.  
  1. “Prostate Cancer Treatment”. Good review of the different techniques and some important issues.  Is it possible to have a paragraph to describe overall indications of focal treatment, the rate of recurrence of each method, the way to diagnosis of the recurrence following focal treatment, and complications? 

Author Response

Dear Cancers Journal Editorial Board, and Reviewers, 

We want to thank you for taking the time to thoroughly review our manuscript and provide excellent comments.  We appreciate the positive feedback and have completed the suggested and necessary changes. Please see revised manuscript with tracked changes and detailed responses below. Reviewer comments are in italics and our responses in bold. 

Sorry for any delay. Thank you again and please contact me for any additional questions. 

Sincerely, 

Seyedeh Nina Masoom, MD 

Reviewer #1: 

Clinically useful review. Text and Tables are clear. Illustrations are mostly reasonable high quality.  

     1. Page 2, line 59, “…detect enhancement…” can be changed to “…detect vivid and early enhancement within PCa…”.    

Thank you for the positive feedback. Changed to “detect vivid and early enhancement within PCa”. 

     2. “Transrectal approach is more commonly used but has a higher risk of sepsis and is also associated with hematochezia. Transperineal approach is increasingly becoming more popular due to lower risk of sepsis. It is, however, associated with a higher rate of urinary retention.” Line 84 to 87. We performed US/MRI fusion biopsy by transrectal approach with infection rate less than 0.5%, very similar to the rate of infection by transperineal approach, because we gave IV antibiotic before the biopsy.  I do not think transperineal approach is becoming more popular.  

Thank you for the valuable comment and sharing the experience in your institution. There is still ongoing discussion regarding the efficacy of these two methods between proceduralists. Most data backing up both approaches have similar PCa detection rate, but majority of studies show increased in risk of sepsis/infection by TR approach, although there are studies indicating no significant difference. We have now added these references to the manuscript. 

However, we do make a statement that some institutions give IV antibiotics to reduce the risk of sepsis after transrectal biopsy which we hope you find agreeable. 

     3. “In-bore Transrectal….” Line 88 to 96. Although the Table 1 mentioned the disadvantages of the method, more details of the disadvantages can be emphasized in this paragraph.  1, time consuming, 40 to 60 minutes per case as compared to the transrectal approach, 20 to 30 minutes. 2, You cannot perform systematic prostate biopsy at the same time of the in-bore.  As we know, up to 10% clinically significant PCa will be missed by prostate MRI.  Imaging guided biopsy together with TRUS biopsy can pick up this 10% PCa.  

We agree with the reviewer comments. We have revised statements in this section to incorporate your comments. 

      4. “Prostate Cancer Treatment”. Good review of the different techniques and some important issues.  Is it possible to have a paragraph to describe overall indications of focal treatment, the rate of recurrence of each method, the way to diagnosis of the recurrence following focal treatment, and complications?  

Per reviewer’s comments we have added a table (Table 3.) summarizing the main principles of focal treatments.  

Regarding rate of recurrence, the data for each procedure is not complete as several clinical trials are ongoing.    

Regarding diagnosis of recurrence following focal treatment, we include new statements on diagnosis of recurrence following focal treatment in the discussion.  

Regarding complications, we include a separate paragraph under section 3.5.. 

Reviewer 2 Report

The authors propose a review of PCa MR guided interventions. The topic is interesting and the manuscript plot is comprehensive and complete 

Author Response

Dear Cancers Journal Editorial Board, and Reviewers, 

We want to thank you for taking the time to thoroughly review our manuscript and provide excellent comments.  We appreciate the positive feedback and have completed the suggested and necessary changes. Please see revised manuscript with tracked changes and detailed responses below. Reviewer comments are in italics and our responses in bold. 

Sorry for any delay. Thank you again and please contact me for any additional questions. 

Sincerely, 

Seyedeh Nina Masoom, MD 

Reviewere #2 

The authors propose a review of PCa MR guided interventions. The topic is interesting, and the manuscript plot is comprehensive and complete. 

Thank you for the comment and positive feedback. 

Reviewer 3 Report

This is a very interesting and well-written review covering a still emerging and current topic. The manuscript was pleasant to read and comprehensive, with good quality figures and useful tables. Congrats on the work well done. I only have two minor suggestions:

1) While photodynamic therapy is technically TRUS-guided, treatment planning is performed on MR images and post-treatment changes are best studied with MRI. Thus, have the Authors considered mentioning this focal treatment approach too? Could it deserve to be included at least in the 3.4 section?

2) It is the opinion of this Reviewer that an additional table summarizing focal treatment strategy, basic principle, energy source, main approaches,  image guidance characteristics and so on would be beneficial for the readers.

Author Response

Dear Cancers Journal Editorial Board, and Reviewers, 

We want to thank you for taking the time to thoroughly review our manuscript and provide excellent comments.  We appreciate the positive feedback and have completed the suggested and necessary changes. Please see revised manuscript with tracked changes and detailed responses below. Reviewer comments are in italics and our responses in bold. 

Sorry for any delay. Thank you again and please contact me for any additional questions. 

Sincerely, 

Seyedeh Nina Masoom, MD 

Reviewer #3 

This is a very interesting and well-written review covering a still emerging and current topic. The manuscript was pleasant to read and comprehensive, with good quality figures and useful tables. Congrats on the work well done. I only have two minor suggestions: 

We appreciate the comments and valuable suggestions and thank you for the positive feedback. 

  1. While photodynamic therapy is technically TRUS-guided, treatment planning is performed on MR images and post-treatment changes are best studied with MRI. Thus, have the Authors considered mentioning this focal treatment approach too? Could it deserve to be included at least in the 3.4 section? 

We appreciate the valuable suggestion of discussing photodynamic therapy (PDT) in more depth in our manuscript. Based on reviewer’s comment, we have added a section dedicated to PDT (under section 3.4.). 

    2. It is the opinion of this Reviewer that an additional table summarizing focal treatment strategy, basic principle, energy source, main approaches, image guidance characteristics and so on would be beneficial for the readers. 

Thank you for pointing out the need for adding a more detailed summery of focal treatment strategies. Based on the reviewer’s comment, we have added a table summarizing these treatments (Table 3.). 
